DATA RELEASE

# Closely related, yet phenotypically different - Genome assemblies of two sister species of widow spiders: *Latrodectus hasselti* and *L. katipo*, Theridiidae

Vladislav Ivanov[1,2,*], Kardelen Özgün Uludağ[3], Jutta M. Schneider[3], Yannis Schöneberg[1], Susan Kennedy[1], Alexander Ben Hamadou[4], Cor J. Vink[5] and Henrik Krehenwinkel[1]

1 Department of Biogeography, Trier University, Universitätsring 15, 54296 Trier, Germany
2 University of Rostock, Mathematisch-Naturwissenschaftliche Fakultät, Institut für Biowissenschaften, Albert-Einstein-Straße 3, 18059 Rostock, Germany
3 Department of Biology, University of Hamburg, Martin-Luther-King Platz 3, D-20146 Hamburg, Germany
4 Senckenberg Research Institute, Senckenberganlage 25, 60325, Frankfurt, Germany
5 Department of Pest-Management and Conservation, Lincoln University, PO Box 85084, 7647, Lincoln, Canterbury, New Zealand

## ABSTRACT

Widow spiders of the genus *Latrodectus* are important animals for biomedical, pest and conservation research. Here, we present the assembled genomes of two closely related *Latrodectus* species: the Australian *L. hasselti* and the New Zealand endemic *L. katipo*. The genome of *L. katipo* consists of 13 scaffolds likely corresponding to chromosomes (90% of the total length) and 1267 short scaffolds (10%). It has a total length of 1.5 Gbp and BUSCO of 94.9%. The genome of *L. hasselti* consists of 379 scaffolds and has a total length of 1.7 Gbp and a BUSCO score of 95.4%. The repeat content is very similar in both genomes with a total proportion of 37.2% for *L. katipo* and 39.9% for *L. hasselti*. Genome annotation predicted 12706 and 15111 genes for *L. katipo* and *L. hasselti* respectively. An ortholog analysis shows large overlap between orthogroups suggesting either duplication events in *L. hasselti* or loss of genes in *L. katipo*.

**Submitted:** 01 March 2026

\* Corresponding author. Email: vladislav.ivanov@uni-rostock.de

Preprint submitted at https://doi.org/10.64898/2026.04.17.719154

**Subjects** Genetics and Genomics, Evolutionary Biology, Animal Genetics

## INTRODUCTION

The widow spiders of the genus *Latrodectus* are widely distributed throughout the world. These spiders are well known for their mating system, ecology, and medical importance. *Latrodectus* spiders possess a potent venom [1] and due to their often synanthropic lifestyle there are regular encounters with humans frequently, requiring medical attention [2, 3]. The common name of "widow spiders" is based on the interesting mating behavior of some species that often leads to death of the male [4, 5]. In the well-studied *L. hasselti* and *L. geometricus*, males perform a spectacular somersault during mating which brings their abdomen onto the female fangs, which results in the male being eaten by the female [4, 6]. However, males can survive their first copulation due to an abdominal constriction which

delays the effects of the female bite [7]. These males then return to inseminate the 2nd spermatheca of the female after which they will not survive. The males are monogynous, a mating system which contradicts conventional sex roles [8].

The majority of other widow spiders including the sister species to *L. hasselti*, the New Zealand endemic katipo spider (*L. katipo*), lack the above mating traits [6]. The two closely related species hybridize and provide an excellent system to study the genetic basis of complex, genetically regulated behaviors [7]. Fromhage et al. (2005) [8] proposed that monogyny can evolve under a male-biased adult sex ratio. Such a bias can arise if males mature faster and hence reach maturity with a higher probability than females. The difference in survival probability will increase with an increased sexual size difference (SSD). Indeed, SSD is less pronounced in *L. katipo* than in *L. hasselti*, presumably causing a stronger sex ratio bias than in *L. katipo* [9].

The invasive *L. hasselti* has arrived in New Zealand [10] and threatens the endemic katipo due to unidirectional introgression. Curiously, hybridization occurs in spite of the differences in mating behavior and in SSD. Mating, however, only works in one direction [11], with *L. hasselti* males mating with *L. katipo* females. This has led to introgression of redback DNA into the gene pool of the katipo [12]. To explore the effect of this introgression, which could threaten the endangered katipo spider, genome data are required.

As a base for future investigations into the complex mating behavior and its consequences, we present a high-quality reference genome for *L. hasselti*, and a chromosome level genome for *L. katipo*. These genomes provide highly valuable resources to study the genetic basis of complex mating traits like self-sacrificial somersault, and to understand the extent of introgressive hybridization on the genetic integrity of katipos in New Zealand.

## MATERIALS AND METHODS

### Specimen sampling

All individuals of *L. katipo* used in the study hatched and were raised in the laboratory at the Department of Biology, University of Hamburg (see Table S1 for metadata, Supplementary Material, SM). The original egg sacs were collected in New Zealand and sent to Germany (Permit 87449-RES). We used two mature females from the first, field collected generation for reference genome generation. Two male specimens were used to obtain DNA for short read sequencing which was necessary to predict sex chromosomes (see below). We also processed 18 male specimens for RNA sequencing. All males were raised in the laboratory in Hamburg. The total number of individual *L. katipo* used in the study is 22.

The sampling of *L. hasselti* was identical except that it was complemented by additional four females coming from Dr Maydianne Andrade lab at University of Toronto (Table S1). The RNA sequencing for the latter samples was done in 2015. The total number of individual *L. hasselti* spiders used in this study is 23.

### DNA extraction

Genomic DNA for females of both species (two *L. katipo* and one *L. hasselti*) used for long read as well as short read sequencing for *L. hasselti* was isolated using Monarch HMW DNA Extraction Kit for Tissue v2.1_4/21 (New England Biolabs). We used only the prosoma and legs of the specimens to reduce contamination originating from the digestive tract which is



mostly located in the abdomen. To avoid potential interference of venom components with further lab work and sequencing, we removed the chelicera which contain most of the venom glands. The animals were put into liquid nitrogen, dissected and the remaining tissue was crushed with a pestle. The DNA isolation was performed following the manufacturer protocol. Two *L. katipo* males used for short read sequencing were put in ethanol in Hamburg and shipped to Trier for DNA isolation. The males were removed from ethanol, left to dry for at least 30 min and then cut into pieces with sterile blade. We used the whole prosoma and all legs. We extended lysis to 16 hours. Otherwise, we followed standard input instructions for QIAGEN Puregene protocol.

## DNA sequencing

After DNA extraction, we prepared a continuous long-read (CLR) sequencing library for *L. katipo* using the SMRTbell Express Template Prep Kit 2.0 (Pacific Biosciences). The CLR data were generated on a PacBio Sequel IIe at the Radboud University Medical Center, Netherlands. The remaining DNA isolate was sent to Novogene for 150 bp paired end sequencing on a NovaSeq X Plus at the target depth of 40× (genome size was derived from the draft assembly) short reads for short read polishing. For *L. hasselti*, we prepared a Circular Consensus Sequencing (CCS) library following the protocol for the whole genome and metagenome libraries using SMRTbell prep kit 3.0. The library was sequenced on the PacBio Revio platform. Both libraries were evaluated using an Agilent Tapestation 2200 and a Quantos Fluorometer.

The second *L. katipo* specimen was used for Hi-C library preparation using the Arima Genomics High Coverage Hi-C Kit following the User Guide for Animal Tissues and the Arima Hi-C+ Kit User Guide for Library Preparation with the Arima Library Prep Module. The library was indexed with the Accel-NGS 2S Plus DNA library kit (Swift Biosciences). Library QC was performed using the Quantos Fluorometer and the Agilent Tapestation 2200. The libraries were sent to Novogene for 150bp paired end sequencing with 60× target read coverage on an Illumina NovaSeq 6000.

To be able to identify the sex chromosomes, we mapped short read sequenced of two *L. katipo* males. DNA-extractions were sent to Novogene for library preparation and sequencing with 15× target read coverage on an Illumina NovaSeq 6000.

## Genome assembly and repetitive sequence content

All programs and commands used for assembly and analysis steps are listed in SM. The raw CLR, Hi-C, short Illumina (*L. katipo*) and CCS (*L. hasselti*) reads were assessed for quality using FastQC v0.12.1 [13]. The genome of *L. katipo* was assembled with wtdbg2 v2.5 [14]. This and consecutive assemblies were assessed for their quality using BUSCO v5.5.0 [15] against arachnida_odb10 and araneae_odb12 [16] and Quast v5.2.0 [17]. The initial *L. katipo* assembly was polished with long (Flye v2.9.2 [18]) and short (Polypolish v0.6.0 [19]) reads. Duplicated contigs were purged with Purge_Dups v1.2.6 [20].

The Hi-C reads were mapped using the Arima Mapping Pipeline A160156 v02 (see SM for details) and the read groups were added using the Picard v3.1.0 command AddOrReplaceReadGroups [21]. Scaffolding was performed with YaHS v1.2a.1.patch [22] and juicer_tools v1.22.01 [23]. Gaps were closed using TGS-GapCloser v1.2.1 [24]. We used BlobToolKit v4.4.3 [25] to identify and remove contaminating sequences by taxonomic assignment. To do so, we performed a blastn v2.14.0 search [26] of the assembly against the

NCBI nucleotide database (downloaded on 24.01.2022) and mapped the reads used for assembly and polishing against the respective assembly using minimap2 v2.26 [27]. We extracted the scaffolds that matched Arthropoda or without a match and these scaffolds comprised the final assembly. The quality was assessed using BUSCO v5.5.0 and Quast v5.2.0. The scaffolds were visualized in Juicebox v2.20.00 [23]. To identify sex chromosomes, trimmed (fastp v0.24.0 [28]) short reads from two *L. katipo* males and one female were mapped against the final assembly (bwa-mem2 v2.2.1 [29]), alignment statistics was calculated with Qualimap v2.3 [30]. As sex chromosomes in males are expected to be haploid, we were able to identify them by having approximately half of the female coverage.

The assembly of the *L. hasselti* genome was done using HiFiasm v0.19.8 [31]. The polishing steps were omitted because CCS data is of much higher quality than CLR. Gap closing, deduplication, and contamination check followed the same pipeline as for *L. katipo*. For cross-species comparison, *L. hasselti* contigs were mapped against the 13 largest *L. katipo* scaffolds using Jupiter plot v1.1 [32].

As part of the genome annotation, we ran a *de novo* repeat identification using RepeatModeler v2.0.5 [33] to generate custom repeat libraries for each genome. Both custom repeat libraries were combined with the invertebrate Repbase v28.08 [34] into a single library, which we used for repeat masking with RepeatMasker v4.1.7-p1 [35] for both genomes. The repeat landscape was created using RepeatMasker's calcDivergenceFromAlign.pl script and the results were plotted using ggplot2 v4.0.1 [36] in R v4.5.0. The set of shared repetitive elements between the species was created by comparing the list of repeats annotated from the reference library. The comparison results were plotted in R using ggplot2 v4.0.1. For more details see SM.

## RNA isolation, sequencing and quality trimming

RNA was extracted from 18 males per species (i.e, 6 subadults, 6 unmated adults, 6 mated adults) using the RNeasy Mini kit (Qiagen, Hilden, Germany). Whole body was used for RNA isolation. Samples were homogenized by bead beating in 350 µL Buffer RLT and the lysate was centrifuged at maximum speed for 3 min. The supernatant was pipetted for the next step. The kit instructions were followed from step 2 onwards. RNA quality and quantity were assessed using the High Sensitivity RNA ScreenTape on a 4200 TapeStation (both Agilent Technologies, Santa Clara, CA, USA).

We prepared two pools per species, each containing RNA from nine individual males (3 subadults, 3 unmated adults and 3 mated adults). From each sample, 50 ng of RNA was added to their corresponding pool. If the total volume of the pool was less than 16 µL, RNase-free water was added until reaching 16 µL of total volume to meet the criteria of sequencing facility (≥10 µL, ≥20 ng/µL).

All samples were sent to NOVOGENE, Munich Germany, for mRNA enrichment via poly-A selection and sequencing. Libraries were 150 bp paired-end sequenced on an Illumina NovaSeq X Plus platform, targeting 9 Gbp per sample. Quality trimming was performed with Trimmomatic version 0.38.1 [37] on the Galaxy server (version 21.09) of the Department of Biology, University of Hamburg. Adapter and other Illumina-specific sequences were removed using the TruSeq3 (paired-end, for MiSeq and HiSeq) dataset. The first 15 nucleotides from the 5′ ends of the raw reads were cropped. Reads shorter than 20 nucleotides were discarded, and only reads with a mean quality score of ≥20 were retained for subsequent analyses.



### Genome annotation and repeat analysis

The soft masked genomes were annotated using BRAKER v3.0.8 [38] with default settings. To facilitate gene prediction, we provided known Arthropoda genes from orthodb 11 database and arachnida_odb10 BUSCO lineages [16]. For *L. katipo*, gene models were trained based on the RNA sequencing data from 18 specimens. In case of *L. hasselti*, RNA reads from the 18 *L. hasselti* specimens sequenced for this study and four females from Universit of Toronto were added as additional evidence for gene prediction.

The resulting amino acid sequences for each species were uploaded to the online version of eggnog-mapper [39] to generate a functional annotation of the predicted genes. The analysis was performed using default parameters except for Taxonomic Scope which was set to Arthropoda. We also ran BUSCO on the predicted amino acid sequences using the arachnida_odb10 dataset.

### Orthofinder analysis

In order to obtain additional insight into differences between gene orthologs in *L. hasselti* and *L. katipo* we compared sequences predicted by BRAKER3 to annotations of nine spider genomes. Ortholog comparison between the species allows to detect duplications, gene loss and provides a groundwork for speciation research. The set of additional species was chosen based on a) genome and annotation availability; b) phylogenetic relationship to the focal species. We used phylogenetic tree of all spiders [40] as guide to select the families so we could include closely related species, related families and an outgroup family. There were three closely related species from the same family (Theridiidae) with genomes and annotation available: *L. elegans* [41], *L. hesperus* [42] and *Parasteatoda tepidariorum* [43]. Representatives of more distantly related families included: Araneidae, *Argiope briennichi* [44], *Nephila* (*Trichonephila*) *pilipes* (GCA_019974015.1); Linyphiidae, *Oedothorax gibbosus* [45]; Pimoidae, *Pimoa clavata* [46] and Tetragnathidae, *Tetragnatha kauaiensis* (GCA_947070885.1). As outgroup, we chose the species *Stegodyphus dumicola* [47] which belongs to the family Eresidae. Accessions are available in bioinformatics section of SM. For *L. elegans*, we had to extract protein sequences file from the gff3 file using gffread v0.12.8 [48] and the respective reference genome.

Next, we selected the longest of the possible transcripts in our own amino acid sequences files for both focal species (see SM) to avoid inflation of duplication event counts. Finally, we ran OrthoFinder v3.0.1b1 [49] for all 11 species with default parameters.

## RESULTS AND DISCUSSION

### Genome sequencing and assembly metrics

DNA sequencing produced 117 Gbp of CLR and 65.6 Gpb of CCS data that corresponds to 73× and 41× coverage of *L. katipo* and *L. hasselti* genomes respectively (Table 1, Figure 1). The final genome assembly of *L. katipo* consists of 1280 scaffolds with a total length of 1.53 Gbp. The longest 13 scaffolds account for 89.5% (1.37 Gbp) of the total assembly length and likely represent the individual chromosomes (Figure 1b). The *L. hasselti* assembly is 1.7 Gbp long and consists of 379 contigs (Table 1). The respective N50 for *L. katipo* and *L. hasselti* are 109 Mbp and 13.4 Mbp, and L50 are six and 33. The majority of contaminating sequences removed from both genomes belonged to Pseudomonadota (Figure S1 and S2, SM).

We identified two scaffolds in the *L. katipo* assembly (scf6 and scf7) as X chromosomes using read coverage. *Latrodectus* species usually exhibit a $X_1X_2O$ sex determination system



**Table 1.** Raw data and genome assembly statistics.

| *L. katipo* | |
|---|---|
| Type of LR | CLR |
| Platform for LR sequencing | Sequel II |
| LR raw, size, Gbp | 548.56 |
| LR coverage for ~1.6 Gbp genome | 342.85 |
| LR, n reads, millions | 6.6 |
| LR, mean length, bp | 82621 |
| Illumina SR for HiC, 150PE, size, Gbp | 84.5 |
| Illumina SR for HiC, 150PE, number reads millions | 563.6 |
| Illumina SR for polishing, 150PE, size, Gbp | 20.1 |
| Illumina SR for polishing, 150PE, number reads millions | 134 |
| Genome size, Gbp | 1.531 |
| Number of scaffolds | 1280 |
| N50, Mbp | 109 |
| L50, number of scaffolds | 6 |
| BUSCO arachnida_odb 10 (2934 genes), % | 97.7 |
| BUSCO araneae_odb12 (3974 genes), % | 94.9 |
| *L. hasselti* | |
| Type of LR | HiFi |
| Platform for LR sequencing | Revio |
| LR raw, size, Gbp | 65.6 |
| LR coverage for ~1.6 Gbp genome | 41 |
| LR, number reads, millions | 4.5 |
| LR, mean length, bp | 14474 |
| Genome size, Gbp | 1.696 |
| Number of contigs | 379 |
| N50, Mbp | 13.4 |
| L50, number of contigs | 33 |
| BUSCO arachnida_odb 10 (2934 genes), % | 98.6 |
| BUSCO araneae_odb12 (3974 genes), % | 95.4 |

LR - long reads; SR - short reads.

[41, 50, 51]. As sex chromosomes are expected to be haploid for males, their read coverage should be half of the autosomes. The putative autosomes of male *L. katipo* specimens had a short read coverage of 13.5×, whereas the X chromosomes had 6.5× and 6.6×. On the other hand, the average coverage for autosomes (21.2) all chromosomes was very close to X chromosomes' coverage (18.8 and 19.4) in the female *L. katipo* used for short read polishing. The majority of *L. hasselti* contigs (86.5 %) were successfully mapped against the *L. katipo* assembly, representing 99.3 % of the total length of the *L. hasselti* assembly (Figure 1d and Figure S3, SM). The BUSCO scores were different depending on the odb used but for both species assemblies the complete BUSCO percentage was over 90% (Table 1, Table S2 and Figure S4, SM) except for *L. katipo* assembly that included only 13 largest scaffolds and was analyzed against araneae_odb_12.

In the context of already published *Latrodectus* genomes, the assembly of *L. katipo* differs in the number of chromosomes. Both *L. elegans* and *L. hesperus* have 14, while *L. katipo* has 13, and it is very likely that the *L. hasselti* genome consists of 13 chromosomes as well. At the same time, the size of both new genomes is comparable to other *Latrodectus* species [41, 42].



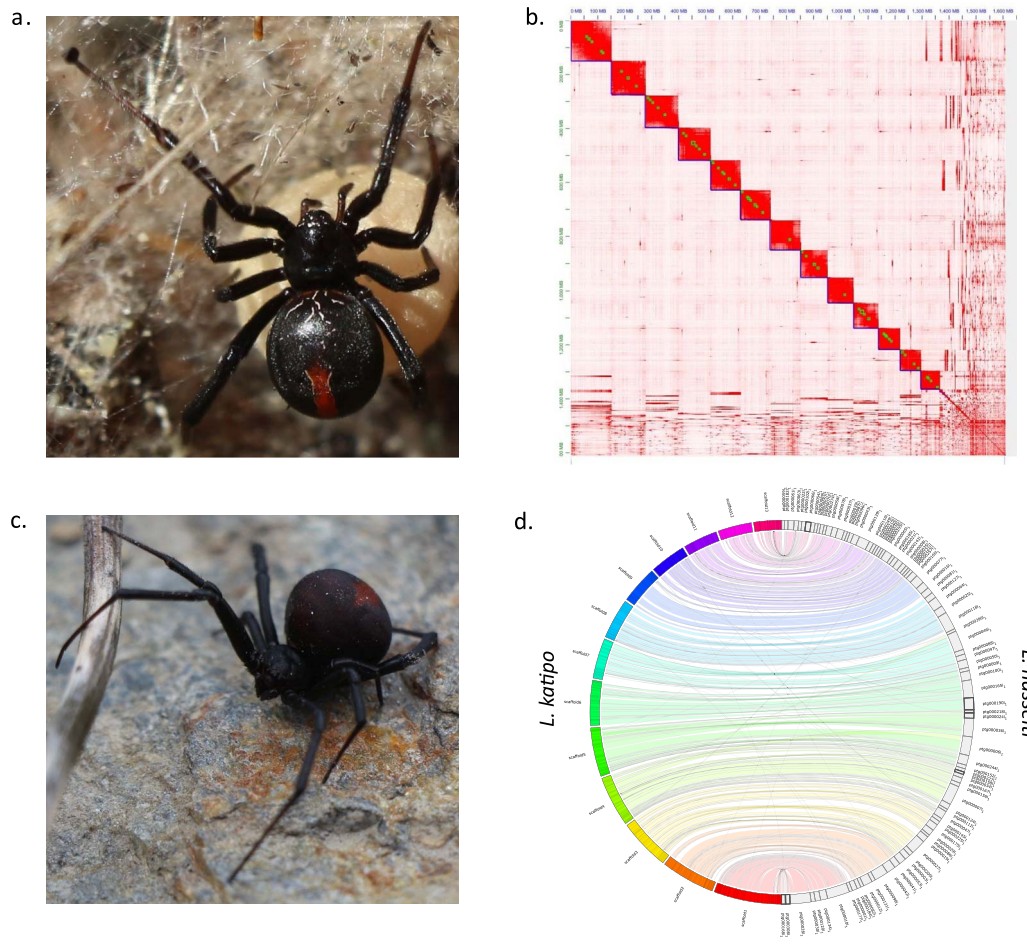

**Figure 1.** (a) Female of *L. katipo*, (b) contact map for scaffolded contigs of *L. katipo*, (c) female of *L. hasselti*, (d) synteny plot for 13 longest scaffolds of *L. katipo* and all contigs of *L. hasselti*.

## Genome annotation and ortholog analysis

The most striking difference between the species is the number of predicted and annotated genes where *L. hasselti* has 19% more genes than *L. katipo*. BRAKER3 predicted 12706 genes and 16196 amino acid sequences for *L. katipo* and 15111 genes and 20289 amino acid sequences for *L. hasselti*. The contigs of *L. hasselti* mapped to 13 *L. katipo* scaffolds harbor the overwhelming majority of the predicted genes (15107). Therefore, the excess is likely not caused by assembly artefacts or contamination. Neither does it seem to be caused by a simple chromosome loss in *L. katipo*. BUSCO scores for the predicted amino acid sequences are 90.7% for *L. katipo* and 98.9% for *L. hasselti* (Table S2 and Figure S4, SM). Lower value for *L. katipo* is likely because gene prediction was done only for 13 largest scaffolds.

The functional assignment (eggNOG orthologs) annotated almost the same proportion of the predicted genes for both species: *L. katipo* 69% and *L. hasselti* 68%. We compared Clusters of Orthologous Genes (COG) categories counts between the species based on their functional annotation. For *L. hasselti* we used only genes located on the contigs that mapped against *L. katipo*. There are 8518 genes with COG categories (124 unique combinations) for *L. katipo* and 9994 genes (122 unique combinations) for *L. hasselti*, i.e.

**Table 2.** Summary statistics of Orthofinder analysis for *L. hasselti* and *L. katipo*.

|  | *L. hasselti* | *L. katipo* |
|---|---|---|
| # of genes | 15111 | 12706 |
| # of genes in OGs | 14840 | 12469 |
| # of unassigned genes | 271 | 237 |
| % of genes in OGs | 98.2 | 98.1 |
| % of unassigned genes | 1.8 | 1.9 |
| # of OGs | 12246 | 10670 |
| % of OGs containing species | 51.7 | 45 |
| # of species-specific OGs | 6 | 2 |
| # of genes in species-specific OGs | 22 | 4 |
| % of genes in species-specific OGs | 0.1 | 0 |
| # - number, % - percentage, OGs – orthogroups. | | |

there are 17% more genes for the redback spider. The proportion of each major COG category in total counts is almost identical between species with the largest being of Cellular processes and signaling (~37% in both species, Table S3, SM). Each category of Information storage and processing, Metabolism and Poorly characterized accounts for 20–21% of all genes in both species (Table S3, SM). There is no obvious pattern in distribution of general and specific COG categories on different scaffolds and it seems that they are very similar between species (Figure S5 and S6, SM). This suggests conserved set of genes on the corresponding chromosomes.

Orthofinder has classified genes from 11 spider species into 23690 orthogroups (OGs), of which 3858 OGs are shared among all analyzed spiders (Table 2 and Table S4, SM). Almost all genes of *L. hasselti* (98.2%) and *L. katipo* (98.1%) were assigned to OGs (Table 2). The inferred species tree is correct with 100% bootstrap support for all nodes (Figure S7). The duplication analysis suggests 268 duplicated genes for *L. katipo* and 750 for *L. hasselti* (Figure S7).

In total, there are 10670 OGs in *L. katipo* and 12246 OGs in *L. hasselti*. For comparison, both *L. katipo* and *L. hasselti* have more OGs than two other *Latrodectus* species despite the fact that the input fasta files for all other species were not stringently filtered, therefore had thousands more amino acid sequences than the focal taxa (Table S4). Difference in gene numbers is also reflected in number of non-overlapping OGs between two species. There are 415 exclusive OGs for *L. katipo* and 1469 OGS for *L. hasselti* (including unassigned OGs, i.e. single genes lacking orthology). Duplicated genes can play an important role in adaptation in spiders as it was shown on *Heteropoda venatoria* [52] and *Octonoba sinensis* [53].

## Repetitive elements

Repeats comprise 37.29% of the *L. katipo* assembly (35.49% if only the 13 chromosomes are considered) and 39.92% of the *L. hasselti* assembly (Table 3). In both cases the majority of transposable elements (TEs) are interspersed repeats and a large proportion is unclassified. The classified repeat content in both species has almost equal proportions of transposons and a very close proportion of retroelements (Table 3) as well as similar repeat landscapes (Figure S8 and S9).

The proportion of unique TEs (including unclassified) that are shared between species (15469) to total number of unique TEs in a genome is 85% for *L. katipo* (18163 total) and 84% for *L. hasselti* (18418 total). The shared TEs seem to have been active at approximately the



**Table 3.** Repeat content report.

| Category | L. katipo | L. hasselti |
|---|---|---|
| Total assembly length, bp | 1530589153 | 1696211549 |
| GC content, % | 27.81 | 28.06 |
| Repeat content total, % | 37.29 | 39.92 |
| Retroelements total, % | 8.16 | 8.19 |
| DNA transposons total, % | 10.54 | 10.34 |
| Rolling-circles, % | 0.05 | 0.05 |
| Unclassified repeats, % | 15.03 | 18.09 |
| Total interspersed repeats, % | 33.74 | 36.63 |
| Small RNA, % | 0 | 0 |
| Satellites, % | 0 | 0 |
| Simple repeats, % | 3.07 | 2.84 |
| Low complexity, % | 0.42 | 0.4 |

same times (Figure S10) but not without outliers. The largest proportion of non-shared TEs belong to the repeats that were active in the past but not the oldest ones (Figure S11 and S12), while the shared TEs are active now or belong to ancient bursts of activity that could have occurred long before the species split. Interestingly, the most active non-shared repeats of *L. hasselti* comprise roughly 80% of the total length of the most active TEs (Figure S13), while *L. katipo* shared TEs account for almost the total length of TEs in the species' genome (Figure S14).

## CONCLUSION

*Latrodectus hasselti* and *L. katipo* are sister species showing intriguing and complex trait differences despite recent divergence, which raises wide reaching questions for speciation, local adaptation and conservation. Their striking differences in traits related to mating involving male self-sacrifice and the possibility of interspecific hybridization make these species promising subjects to identify the genetic pathways controlling this behavior. In the light of the recent invasion of *L. hasselti* into New Zealand, this behavior has important implications for conservation of *L. katipo* and provides interesting opportunities to study speciation mechanisms and adaptive introgression. Here, we present the reference genomes for both species. The genomes are of high quality and the one from *L. katipo* reached chromosome level. We were able to identify sex-chromosomes and provide high quality de-novo assemblies. Therefore, we provide a strong data basis for future studies examining the genetic behavior of complex behavioral traits as well as conservation biology and speciation.

## DATA AVAILABILITY

The raw data is available on NCBI. Bioproject accession for *L. katipo* is PRJNA1250404, bioproject accession for *L. hasselti* is PRJNA1238270. The raw RNA sequences for pooled individuals are: *L. katipo* BioProject PRJNA1274353 and *L. hasselti* BioProject PRJNA1272815.

Assembled genomes, masked genomes, genome annotations and accompanying files, additional figures and tables, Orthofinder results and bioinformatic pipelines and commands are uploaded to Zenodo [54, 55]. The final versions of genomes can be accessed on NCBI: *L. katipo* assembly ASM5682273v1 and *L. hasselti* assembly ASM5682221v1.

## DECLARATIONS

### Ethics approval and consent to participate

The authors declare that ethical approval was not required for this type of research.

### Competing interests

The authors declare that they have no competing interests.

### Funding

The project was funded by German Research Foundation (DFG), project number 458109219 to HK & JMS. PacBio sequencing of *L. katipo* was supported by grants from the DFG in the framework of the priority program SPP 1991: TAXON-OMICS (project number KE 2647/1-1). This publication was funded by Open Access Publication Fund of the University of Rostock.

### Acknowledgements

We would like to thank the LOEWE Centre for Translational Biodiversity Genomics (LOEWE-TBG), which was supported through the program 'LOEWE-Landes-Offensive zur Entwicklung Wissenschaftlich-ökonomischer Exzellenz' of Hesse's Ministry of Higher Education, Research, and the Arts (HMWK), for support in the lab. We would also like to thank the Bioscientia Institut für Medizinische Diagnostik GmbH for providing the PacBio SMRT sequencing service on the PacBio Revio platform. The spiders were collected under the permit number 87449-RES issued by the Department of Conservation, New Zealand.

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
