## [Editor Report]

Editor’s AssessmentThe manuscript is ready for formal acceptance.Editor’s AssessmentThe manuscript is ready for formal acceptance.

---

## [Reviewer Report]

Reviewer name and names of any other individual's who aided in reviewer Guanliang MengDo you understand and agree to our policy of having open and named reviews, and having your review included with the published papers. (If no, please inform the editor that you cannot review this manuscript.)YesIs the language of sufficient quality?YesPlease add additional comments on language quality to clarify if needed
Are all data available and do they match the descriptions in the paper? YesAdditional CommentsAre the data and metadata consistent with relevant minimum information or reporting standards? See GigaDB checklists for examples <a href="http://gigadb.org/site/guide" target="_blank">http://gigadb.org/site/guide</a>YesAdditional CommentsIs the data acquisition clear, complete and methodologically sound?YesAdditional CommentsIs there sufficient detail in the methods and data-processing steps to allow reproduction?YesAdditional CommentsIs there sufficient data validation and statistical analyses of data quality? YesAdditional CommentsIs the validation suitable for this type of data?YesAdditional CommentsIs there sufficient information for others to reuse this dataset or integrate it with other data?YesAdditional CommentsAny Additional Overall Comments to the AuthorThe authors present the genomes of two widow spiders, Latrodectus hasselti and L. katipo, representing a siginficant contribution to spider genomics. The manuscript was nicely written in general. Line 141: The “40X” (and similar words in the manuscript) assumes we already knew the genome size of the target species, however, this is not explained in the text. Lines 147- 152: Duplicate sentences? Lines 188-196 and Lines 220-228: Duplicate sentences. Line 164: What are the “consecutive assemblies” in the text? Line 165: Please indicate how many genes in the database arachnida_odb10 and araneae_odb12. Line 218: remove the ” in the end of the paragraph. Line 233, 254. Please double-check the namings/numberings of Figures, Tables in the manuscript. Line 280: the mapping rate of 86.5% is a bit low. Should be explained briefly in the result section. Line 295: Please provide the BUSCO plots for both species (could be placed in the SM).RecommendationMinor Revision

---

## [Reviewer Report]

Reviewer name and names of any other individual's who aided in reviewer YimingZhangDo you understand and agree to our policy of having open and named reviews, and having your review included with the published papers. (If no, please inform the editor that you cannot review this manuscript.)YesIs the language of sufficient quality?YesPlease add additional comments on language quality to clarify if needed
The manuscript would benefit from minor language polishing. In particular, some sentences are rather long and complex; breaking them into shorter sentences may improve clarity and enhance the overall readability.Are all data available and do they match the descriptions in the paper? YesAdditional CommentsThe sequencing strategy for L. katipo requires clarification. The main text appears to describe conventional long-read sequencing, while the table indicates CCS data. As CCS reads have substantially higher accuracy and typically do not require short-read polishing, whereas conventional long reads usually do, this distinction is important. Please clarify and ensure consistency throughout the manuscript.Are the data and metadata consistent with relevant minimum information or reporting standards? See GigaDB checklists for examples <a href="http://gigadb.org/site/guide" target="_blank">http://gigadb.org/site/guide</a>YesAdditional CommentsIs the data acquisition clear, complete and methodologically sound?YesAdditional CommentsLine: 147–149. The Hi-C library preparation section appears to contain some repetitive descriptions of the same protocols. Streamlining this part may improve readability. In addition, please verify that the species names are consistent, as this section may contain an unintended reference.Is there sufficient detail in the methods and data-processing steps to allow reproduction?YesAdditional CommentsIs there sufficient data validation and statistical analyses of data quality? YesAdditional CommentsThe authors may consider evaluating additional assemblers (e.g., hifiasm) for contig construction, as different tools can yield improved assembly continuity and accuracy. Furthermore, given that the current Hi-C scaffolding results appear not fully optimal, testing alternative Hi-C scaffolding workflows may help further improve the chromosome-level assembly.Is the validation suitable for this type of data?YesAdditional CommentsIs there sufficient information for others to reuse this dataset or integrate it with other data?YesAdditional CommentsAny Additional Overall Comments to the AuthorThis study provides valuable genomic resources for two closely related widow spider species with distinct mating systems and conservation significance. The availability of these genome assemblies will facilitate future research on behavioral evolution, species divergence, and introgression in Latrodectus. Overall, the dataset represents an important contribution to arachnid genomics and evolutionary biology.RecommendationMinor Revision